# Poly(alkyl-biphenyl pyridinium)-Based Anion Exchange Membranes with Alkyl Side Chains Enable High Anion Permselectivity and Monovalent Ion Flux

**DOI:** 10.3390/membranes13020188

**Published:** 2023-02-03

**Authors:** Jin Yang, Qian Chen, Noor Ul Afsar, Liang Ge, Tongwen Xu

**Affiliations:** 1Anhui Provincial Engineering Laboratory of Functional Membrane Materials and Technology, Department of Applied Chemistry, School of Chemistry and Materials Science, University of Science and Technology of China, Hefei 230026, China; 2Applied Engineering Technology Research Center for Functional Membranes, Institute of Advanced Technology, University of Science and Technology of China, Hefei 230088, China

**Keywords:** anion exchange membrane, monovalent anion permselective membrane, hydrophobicity, electrodialysis, permselectivity

## Abstract

Poly(alkyl-biphenyl pyridinium)-based anion exchange membranes with alkyl side chains were synthesized for permselective anion separation. By altering the length of the grafted side chain, the hydrophilicity and other attributes of the membranes could be controlled. The QDPAB-C5 membrane with the best comprehensive performance exhibited a Cl^−^ ion flux of 3.72 mol m^−2^ h^−1^ and a Cl^−^/SO_4_^2−^ permselectivity of 15, which are significantly better than the commercial Neosepta ACS membrane. The QDPAB-C5 membranes with distinct microscopic phase separation structures formed interconnected hydrophilic/hydrophobic ion channels and exhibited excellent ion flux and permselectivity for other anionic systems (NO_3_^−^/SO_4_^2−^, Br^−^/SO_4_^2−^, F^−^/SO_4_^2−^, NO_3_^−^/Cl^−^, Br^−^/Cl^−^, and F^−^/Cl^−^) as well. Furthermore, the influence of alkyl side chain length on the membranes’ ion flux and permselectivity in electrodialysis was investigated, which may be attributed to the alterations in ion channels and hydrophobic regions of the membranes. This work provides an effective strategy for the development of monovalent anion permselective membranes.

## 1. Introduction

With the expansion of the social economy and industry, the resulting environmental issues have pushed researchers to call for stricter standards in the fields of resource recovery, wastewater treatment, and seawater desalination in order to meet the requirements of a sustainable green economy [1,2]. A typical scenario is the discharge of highly concentrated fluoride, bromide, nitrate, and sulfate byproducts originating from many manufacturing procedures, such as the petroleum, paper, smelting, photovoltaic, and semiconductor industries [3,4]. Arbitrary discharge of these anions into surface water results in toxic effects on the environment and human health. This grim reality has attracted the attention of researchers to treating industrial wastewater and recovering these byproducts. The greatest challenge in recovering resources is efficient separation of the anions existing in industrial wastewater. From this point of view, membrane technologies, such as nanofiltration (NF) and electrodialysis (ED), are generally regarded as the most promising technological means for ion separation due to their low cost and energy consumption, high efficiency, and environment compatibility [5]. In particular, the ED process has a high ion permselectivity and is able to operate at high concentration gradients [6]. With the demand for the separation of monovalent anions by ED, researchers have been motivated to develop anion exchange membranes with high monovalent anion permselectivity.

Numerous studies have reported the fabrication of monovalent anion permselective membranes (MAPMs) by crosslinking, surface modification, and self-assembly methods, according to the separation mechanisms of pore-size screening and electrostatic repulsion [7,8,9,10,11,12]. Surface modification with an oppositely charged layer using electrically driven layer-by-layer assembly technology has led to the improvement in ion permselectivity of the membranes, but the cumbersome preparation process, high area resistance, and instability of the functional layer limit the application of these membranes [13,14,15]. By crosslinking the original anion exchange membrane with a negatively charged polymer or molecule, the permselectivity of MAPMs can be enhanced [16,17,18]. However, due to the compact structure of the polymer membranes, their ion flux and permselectivity have a “trade-off” effect, and the improvement remains restricted. Other researchers have created membranes with novel nanoscale architectures of ion channels that enable efficient permselectivity, whereas the lower ion fluxes of these membranes restrict their widespread industrial applicability [19,20]. In comparison to the other major types of surface-modified membranes, homogeneous anion exchange membranes offer a straightforward preparation procedure, low cost, and excellent stability. As a result, there is a significant market for the development of homogeneous anion exchange membranes with an isotropic structure, high permselectivity, and long-term stability for a wide range of industrial applications.

The hydrophilicity of ion exchange groups in the membrane matrix causes the ion exchange membrane to absorb water. Excessive water uptake will diminish the membrane’s dimensional stability and mechanical strength, while sufficient ion exchange capacity (IEC) is required for high ion flux. Moreover, the difference in Gibbs free energies of anion hydration plays a more important role in ion separation than the difference in ionic radius [21,22]. Previous research has demonstrated that the rate of ion permeation through the membrane with varying Gibbs free energies of hydration can be controlled by altering the hydrophobicity of the membrane [23,24,25,26]. Recently, grafting a side chain to the polymer backbone, which can control lower levels of IEC and simultaneously achieve high permselectivity, ion flux, and dimensional stability, has been proven to be an effective strategy to regulate membrane hydrophobicity [27,28,29]. For example, Irfan et al. developed and synthesized a series of quaternized poly(2,6-dimethylphenyl oxide)-containing alkyl spacers of different chain lengths to achieve a Cl^−^/SO_4_^2−^ permselectivity of 13.1 with a Cl^−^ ion flux of around 1 mol m^−2^ h^−1^. Liao et al. [30]. manipulated the performance of the membranes by adjusting the length of the alkyl spacers grafted to the imidazole-salt-capped side chains of the poly(arylene ether sulfone) backbone to control the performance of the membrane with a Cl^−^/SO_4_^2−^ permselectivity of 7.1 [31]. By constructing an anion exchange membrane with microphase separation morphology, the hydrophilic/hydrophobic microstructure of the membrane can form interconnected ion transport channels. In terms of main-chain materials, polymers containing aromatic pyridines represent an undeveloped field. Heterocycles contribute to the thermal stability, chemical stability, and excellent mechanical strength of polymers. The alkalinity of the nitrogen atoms in heterocycles enables a range of reactions, including quaternization, protonation, and metal complexation [32].

In this study, a poly(alkyl-biphenyl pyridine) polymer was synthesized and then quaternized with bromoalkanes by varying carbon chain lengths to form a dense and uniform anion exchange membrane. The process is conducted at room temperature using a two-step, one-pot method. By changing the length of the extended alkyl spacer, it is anticipated that the hydrophobicity of the prepared anion exchange membranes with varying alkyl spacer lengths will increase in turn. The ion flux and permselectivity of the membranes were assessed using ED experiments on diverse anion systems (NO_3_^−^/SO_4_^2−^, Br^−^/SO_4_^2−^, F^−^/SO_4_^2−^, NO_3_^−^/Cl^−^, Br^−^/Cl^−^, and F^−^/Cl^−^), and the regularity governing the ion flux and permselectivity of these membranes was explored in conjunction with various physical and chemical features of the membranes.

## 2. Materials and Methods

### 2.1. Chemicals and Materials

The chemicals used in the experiment are all of analytical grade. Biphenyl, 4-acetylpyridine, trifluoromethanesulfonic acid (TFSA), and trifluoroacetic acid (TFA) were purchased from Saan chemical technology Co., Ltd., Shanghai, China. Bromoethane, 1-bromopropane, 1-bromopentane, and 1-bromheptane were supplied by Shanghai Aladdin Biochemical Technology Co., Ltd., Shanghai, China. Sinopharm Group Chemical Reagent Co., Ltd., Beijing, China supplied sodium chloride (NaCl), sodium fluoride (NaF), sodium bromide (NaBr), sodium nitrate (NaNO_3_), sodium sulfate (Na_2_SO_4_), potassium nitrate (KNO_3_), N-methyl-2-pyrrolidone (NMP), dichloromethane, and dimethyl sulfoxide (DMSO). The cation exchange membrane (CM-2) and monovalent anion permselective membrane (ACS) were purchased from ASTOM, Tokuyama Soda, Japan.

### 2.2. Preparation of Poly(alkyl-biphenyl pyridine) (PAB)

Typical synthesis procedures of PAB were as follows: biphenyl (6 g), 4-acetylpyridine (6.3 g), and dichloromethane (10 mL) were added to a flask and mechanically agitated for 2 h until thoroughly dissolved. A total of 2 mL of TFA and subsequent 22 mL of TFSA were then added slowly to the above-mentioned system under ice bath conditions. After the feeding, the reaction system continued to stir for 2 h at room temperature and gradually became a purplish-red solution. The purplish-red solution was then slowly poured into deionized water to obtain a yellow-fibrous product. The product was immersed into a 1 mol L^−1^ NaOH solution of for 24 h, filtered, and washed with excess deionized water to make sure the residual acid in the product was neutralized and deprotonation of PAB was complete. Finally, the deprotonated PAB (DPAB) polymer was dried at 40 °C for 24 h.

### 2.3. Quaternization of DPAB and Preparation of Homologous Membranes

The 12.5 wt% DPAB solution was prepared by dissolving 0.01 mol of DPAB in NMP. A total of 0.004 mol of bromoalkanes (bromoethane, 1-bromopropane, 1-bromopentane or 1-bromheptane) was then added into the DPAB solution with mechanical stirring at 60 °C for 24 h to generate a purple-red solution. Each quaternized product was designated as QDPAB-Cx, where x represents the number of carbon atoms of the grafted alkyl side chain. The solution was filtered through a polytetrafluoroethylene filter membrane, cast on a clean glass plate, and dried at 80 °C for 24 h to evaporate the solvent. QDPAB-Cx membranes were then obtained by immersing the glass plate into deionized water and stored in a 0.5 M NaCl solution.

### 2.4. Membrane Characterizations

#### 2.4.1. Structure Characterization

Nuclear magnetic resonance (NMR) spectroscopy (Bruker Ascend 400M, Romanshorn Switzerland) was used to test ^1^H NMR spectra, and DMSO-d6 and CDCl_3_ were utilized as solvents to gain information about the chemical functional groups in the polymer. Attenuated total reflection Fourier transform infrared spectroscopy (ATR-FTIR) (Bruker Vector 22, Romanshorn, Switzerland) was used to characterize the functional groups on the membrane surface (all membranes are dried in an oven at 60 °C for 12 h before measurement). The static water contact angle (CA) of the membranes was measured with a water contact angle meter (DataPhysics OCA20, Stuttgart, Germany) at room temperature using the static drop method.

#### 2.4.2. Ion Exchange Capacity, Water Uptake and Swelling Ratio

The ion exchange capacity (*IEC*) of QDPAB-Cx membranes was calculated via the Mohr method. The membrane in Cl^−^ form was dried at 60 °C for 24 h, weighed, and recorded as *W_dry_* (g). The membrane was then immersed in a 0.3 M Na_2_SO_4_ solution for 24 h for a complete exchange of Cl^−^ ions by SO_4_^2−^ ions. Finally, an automatic potentiometric titrator (Thundermagnetic ZDJ-4B, Shanghai, China) was used to detect the released Cl^−^ ions by titrating with 0.01 mol L^−1^ aqueous solution of AgNO_3_ (*C_AgNO3_*, mol L^−1^). The consumed volume of AgNO_3_ (*V_AgNO_*_3_, L) was recorded. The IEC (mmol g^−1^) of the membrane was calculated using the following formula:(1)$$IEC\;\left(\mathrm{mmol}\cdot g^{-1}\right)=\frac{C_{AgNO_{3}}\times V_{AgNO_{3}}}{W_{dry}}$$

To investigate the dimensional stability of the membrane, the swelling ratio (*SR*) and water uptake (*WU*) were evaluated. The mass difference between the wet and dry membrane samples was used to compute the membrane’s WU. The membrane in Cl^−^ form was immersed in deionized water for 24 h, weighed after gently removing surface water with tissue paper, and recorded as (*W_wet_*, g). Similarly, the swelling ratio (*SR*) was computed using the difference in area between the wet (*S_wet_*, cm^2^) and dry (*S_dry_*, cm^2^) membrane samples.
(2)$$WU\left(\%\right)=\frac{W_{wet}-W_{dry}}{W_{dry}}\times 100\%$$
(3)$$SR\left(\%\right)=\frac{S_{wet}-S_{dry}}{S_{dry}}\times 100\%$$

#### 2.4.3. Membrane Transport Number

The membrane was positioned in the center of the two chambers (Appendix A). Two different concentrations of NaCl (*C_1_* = 0.05 mol/L, *C_2_* = 0.01 mol/L) solutions were then injected into chambers on either side of the membrane. The transmembrane voltage drop across the Ag/AgCl reference electrode was measured using a Keithley multi-output direct current (DC) power supply device at 25 °C. The transport number (*t_i_*) was calculated after recording the open circuit voltage (*E_m_*) data using the modified Nernst equation:(4)$$E_{m}=\left(2t_{i}-1\right)\frac{RT}{zF}\ln(\frac{c_{1}}{c_{2}})$$
where *C_1_* and *C_2_* represent the concentrations of the NaCl solutions in the two separate compartments, *R* represents the constant for an ideal gas (8.314 J mol^−1^ K^−1^), z represents the charge number of counterions (*z* is 1 in this study), and *T* represents the test temperature (K).

#### 2.4.4. Mechanical Properties

The dry membrane was cut into a sample size of 50 mm × 10 mm using a mold. The thickness of each membrane was measured with a screw micrometer. The tensile strength and elongation at break of the dry membrane were measured using a dynamic thermomechanical analyzer (DMA Q-800, New Castle, DE, USA) at a tensile speed of 0.5 N min^−1^.

#### 2.4.5. Current–Voltage (I–V) Curves and Membrane Area Resistance

A four-chamber apparatus (Appendix A) was used to assess the I–V curves of the membranes. A Nafion 115 membrane was used to separate the cathode and anode compartments from the interior compartments. A membrane with an effective area of 4.8 cm^2^ was installed between the two intermediate chambers. At room temperature, the two outer chambers were filled with a 0.3 mol L^−1^ Na_2_SO_4_ solution, while the two inner chambers were filled with a 0.5 mol L^−1^ NaCl solution. During the measurement, a peristaltic pump (LEADFLUID BT600L-2 × YZ15, Baoding, China) with a flow rate of 10 mL min^−1^ was utilized to avoid concentration polarization. A DC power supply (Hansheng Puyuan HSPY-120-01, Beijing, China) was utilized to produce a progressively growing current between two electrodes, which were constructed of ruthenium-coated titanium. A pair of Ag/AgCl reference electrodes were placed in close proximity to the surface of the test membrane to measure the resulting voltage using a multimeter. The area resistance of the membrane can be calculated using the following formula:(5)$$R\left(\Omega \cdot {\mathrm{cm}}^{2}\right)=\frac{U-U_{0}}{I}\times S$$
where *U_0_* represents the voltage drop between the two Ag/AgCl reference electrodes without a membrane.

#### 2.4.6. Electrodialysis Experiment

A four-chamber electrodialysis apparatus, consisting of a cathode chamber, diluted chamber, concentrated chamber, and anode chamber (Appendix A), was used to evaluate the membrane’s separation performance. Two commercial cation exchange membranes (CM-2) were close to the electrode chamber for separating the electrode chamber. The investigated membrane with an effective area of 21 cm^2^ was put in the center to divide the diluted and concentrated chambers. The diluted chamber contained 100 mL of a mixed solution of 0.1 mol L^−1^ NaCl and 0.1 mol L^−1^ Na_2_SO_4_, while the concentrated chamber contained 100 mL of a 0.01 mol L^−1^ KNO_3_ solution. The cathode chamber contained 100 mL of 0.3 mol L^−1^ Na_2_SO_4_ solution and circulated with the anode chamber. The experiment was conducted for 1 h using a regulated DC power supply (Hansheng Puyuan HSPY-120-01, Beijing, China) under a constant current density of 10 mA cm^−2^. After 1 h of running, the concentration of SO_4_^2−^ ions in the concentrated chamber was tested using inductively coupled plasma optical emission spectrometry (ICP-OES) (Optima 7300 DV, Massachusetts, United States), and the concentration of Cl^−^ ions was tested using automatic potentiometric titrator (Thundermagnetic ZDJ-4B, Shanghai, China). The ion flux of each ion can be calculated using the following formula:(6)$$J=\frac{\left(C_{1}-C_{0}\right)V}{A_{m}t}$$
where *C_1_* and *C_0_* represent the initial and final concentrations of Cl^−^ or SO_4_^2−^ ions in the concentrated chamber, respectively, *V* represents the volume of the concentrated chamber, *t* represents the running time, and *A_m_* represents the effective area of the investigated membrane.

The permselectivity between Cl^−^ or SO_4_^2−^ ions was calculated using the following formula:(7)$$P_{SO_{4}^{2-}}^{Cl^{-}}=\frac{J_{Cl^{-}}}{J_{SO_{4}^{2-}}}\times \frac{C_{SO_{4}^{2-}}}{C_{Cl^{-}}}$$
where $J_{Cl^{-}}$ and $J_{SO_{4}^{2-}}$ are the fluxes of Cl^−^ and SO_4_^2−^ ions, respectively. $C_{Cl^{-}}$ and $C_{SO_{4}^{2-}}$ are the initial concentrations of Cl^−^ and SO_4_^2−^ ions in the diluted chamber, respectively.

## 3. Results and Discussion

### 3.1. Chemical Structure Characterization of DPAB and QDPAB-Cx Polymers and the Homologous Membranes

Figure 1a shows the synthesis route of DPAB and QDPAB-C5 polymers. The ^1^H NMR spectra of DPAB and QDPAB-C5 polymers are shown in Figure 1b. In comparison to QDPAB-C5, DPAB only reveals benzene and pyridine characteristic peaks with chemical shifts ranging from 7 to 9 ppm. This is in line with past research [32]. After grafting alkyl group onto the side chain in the quaternization reaction, the spectrum of QDPAB-C5 demonstrates the distinctive peaks of methyl and methylene belonging to the alkyl side chain between chemical shifts of 0.5 ppm and 2.3 ppm. In addition, the ^1^H NMR spectra of QDPAB-C2, QDPAB-C3, QDPAB-C5, and QDPAB-C7 are depicted in Appendix A. These characteristic peaks indicate the successful synthesis of QDPAB-Cx. Figure 1c depicts the ATR-FTIR spectra of DPAB and QDPAB-Cx membranes. After the quaternization reaction, the membranes contain new peaks at 1635 cm^−1^ compared to the DPAB membrane, which is attributable to the production of pyridinium groups after the Menshutkin reaction. The absorption peak at 1595 cm^−1^ is attributable to the pyridine ring’s C=N stretching vibration, and the peak here is diminished to some extent due to the quaternization reaction. The bands located at 2850–2950 cm^−1^ are typical peaks of the alkyl chain’s C–H bond stretching vibration. More methylene is detected as the length of the grafted alkyl chain increases, and the bands become sharper. These findings support the notion that the alkyl side chains were successfully grafted during the quaternization reaction.

### 3.2. Physicochemical Properties of QDPAB-Cx Membranes

The IEC is a critical characteristic that influences ion transport properties. Increasing the IEC assists in enhancing the ion exchange capability of the AEM. As shown in Figure 2a, the IEC of the QDPAB-Cx membranes decreases with the increasing length of the alkyl side chain. Although the same grafting ratio of QDPAB-Cx polymers was designed, the overall quality of QDPAB-Cx polymers increases with an increase in the length of the alkyl side chain, which results in decreasing calculated IEC values. WU and SR are strongly related to IEC, and their trends are similar to those of the IEC. On the other hand, excess WU and SR will cause dimensional instability and poor mechanical characteristics, which is a trade-off effect. The IEC must therefore strike a balance between dimensional stability and ion exchange performance. The introduction of hydrophobic alkyl side chain produces the ion exchange groups and at the same time inhibits the swelling of the membranes to a certain extent. Compared with the commercial Neosepta ACS membrane, all the QDPAB-Cx membranes show a lower WU and SR. In particular, the QDPAB-C7 membrane exhibits extremely low WU and SR (3.8% and 2%, respectively). Furthermore, as illustrated in Figure 2b, the contact angle increases significantly from 62.4° to 79.4° as the length of the grafted alkyl chain increases, and the addition of the long side chain makes the membrane more hydrophobic, which will strongly affect the transport rate of anions with different hydrophobic energies.

The transport number is an essential quantity in ED used to describe the proportion of total current carried by counterions through the membrane. As shown in Figure 2c, the transport numbers of all the QDPAB-Cx membranes are greater than 0.90 and basically equal to that of the commercial Neosepta ACS membrane. Specifically, as the length of the alkyl side chain increases, so does the membrane’s transport number. Generally speaking, the transport number of a membrane varies with the IEC. Due to the hydrophobic nature of chloride ions, they can also permeate the hydrophobic region of the prepared membrane, which is sufficient to compensate for the modest fall in IEC [30].

To meet the requirements of membrane assembly and operation in practical applications, the AEM needs excellent mechanical properties. As shown in Figure 2d, as the length of the grafted side chain increases, the tensile strength of QDPAB-Cx membranes decreases from 43.1 MPa of QDPAB-C2 to 34 MPa of QDPAB-C7, while the elongation at break increases from 12.2% of QDPAB-C2 to 28.9% of QDPAB-C7. This trend is attributed to the introduction of incompatible side chains, which damages the intermolecular interaction with the polymer backbone and destroys the tensile strength of the membrane. This incompatibility can be enhanced by grafting longer alkyl chain branches [31]. On the other hand, the flexible side chains are favorable to the toughness of the membranes, which results in the improvement in elongation at break of the QDPAB-Cx membranes. All in all, QDPAB-Cx membranes exhibit reasonably high mechanical properties, which can meet the requirements of ED applications in the actual process.

### 3.3. Electrochemical Properties of QDPAB-Cx Membranes

I–V curves of QDPAB-Cx membranes and ACS membranes were investigated to evaluate the electrochemical properties of the membranes, as depicted in Figure 3a. A typical I–V curve has an ‘S’-like shape and consists of three distinct regions as follows: The first linear region at a low current density is an ohmic region. Under conditions of low current density, the potential drop increases linearly with the increasing current. In this region, the system resistance mainly depends on ion migration, whereas the area resistance of the membrane is calculated according to formula 5. The slope then decreases and the curve flattens, which is the so-called platform region. At this point, the migration velocity of ions in the membrane will exceed their diffusion rate in the solution, and the concentration of ions near the interface will rapidly decrease, resulting in an increase in area resistance. The diffusion limit value is referred to as the limiting current density, which is calculated by the intersection of the tangent lines of the ohmic region and platform region (Figure 3b). As the voltage continues to increase, the change in voltage will diminish, and this region is called the electric convection region. At this time, water splitting and electric convection will significantly reduce membrane and electrodialysis performance [33]. Therefore, the applied current density during electrodialysis must be less than the limiting current density to prevent concentration polarization, which would result in a decrease in electrodialysis efficiency and an increase in energy consumption.

According to the calculation results, the order of area resistance of the membranes is QDPAB-C2 (2.1 Ω cm^−2^) < QDPAB-C3 (2.2 Ω cm^−2^) < QDPAB-C5 (3.6 Ω cm^−2^) < ACS (4.3 Ω cm^−2^) < QDPAB-C7 (9.1 Ω cm^−2^) (Figure 3c); the order of limiting current density is QDPAB-C2 (108.3 mA cm^−2^) > QDPAB-C3 (100 mA cm^−2^) > QDPAB-C5 (99 mA cm^−2^) > ACS (91.6 mA cm^−2^) > QDPAB-C7 (75 mA cm^−2^). Except for the QDPAB-C7 membrane, the area resistance and limiting current density performance of the other membranes are better than that of the commercial Neosepta ACS membrane, indicating that the membrane can meet the application requirements under normal conditions. The limiting current density is closely related to the membrane’s area resistance. With the increase in alkyl side chain length, the area resistance of the membrane increases and the limiting current density decreases. This could be related to the corresponding AEM’s IEC. High IEC indicates that the AEM membrane has a stronger ion exchange capacity and hydrophilicity, which usually leads to lower area resistance and a higher limiting current density.

### 3.4. Evaluation and Comparison of Anion Permselectivity

As shown in Figure 4a, as the length of the grafted alkyl side chain increases, the Cl^−^ ion flux increases from 3.1 mol m^−2^ h^−1^ of QDPAB-C2 to 3.8 mol m^−2^ h^−1^ of QDPAB-C7, while the SO_4_^2−^ ion flux decreases from 0.36 mol m^−2^ h^−1^ of QDPAB-C2 to 0.242 mol m^−2^ h^−1^. There are two main reasons for this. On the one hand, due to the increased immiscibility between hydrophobic and hydrophilic segments in the membrane matrix, self-assembled ion clusters are compelled to form interconnected hydrophilic channels that serve as a highway for anion transport. As the length of the grafted alkyl side chain increases, when the membrane has an appropriately narrow ion channel, the small Cl^−^ ion is more likely to occupy the ion channels than the large SO_4_^2−^ ion, as shown by the increase in Cl^−^ ion flux and the decrease in SO_4_^2−^ ion flux, resulting in increased permselectivity. On the other hand, as the proportion of hydrophobic methylene in the membrane rises, the membrane’s hydrophobicity increases, as demonstrated by the changes in WU and contact angle. Cl^−^ and SO_4_^2−^ ions have Gibbs free energies of hydration of −317 kJ/mol and −1000 kJ/mol (Table 1) [34], respectively. It is difficult for SO_4_^2−^ ion with larger Gibbs free energy of hydration to pass through the hydrophobic membrane, but the Cl^−^ ion with its low Gibbs free energy of hydration can easily pass through the membrane, which is crucial for anion flux and permselectivity. Similar results were reported by Li et al. [33] and Wang et al. [5]. As a result, QDPAB-C7 exhibited the highest Cl^−^ ion flux of 3.8 mol m^−2^ h^−1^ and Cl^−^/SO_4_^2−^ permselectivity of 15.7, compared with QDPAB-C2, QDPAB-C3, and QDPAB-C5 membranes, which were significantly greater than those of commercial Neosepta ACS membranes (Cl^−^ ion flux of 2.75 mol m^−2^ h^−1^ and Cl^−^/SO_4_^2−^ permselectivity of 6.8).

After comprehensive consideration of area resistance, anion flux and permselectivity of QDPAB-Cx membranes, the QDPAB-C5 membrane was selected as a representative for further investigation. As shown in Figure 4b, the QDPAB-C5 membrane also exhibits high anion flux and permselectivity when applied in NO_3_^−^/SO_4_^2−^, Br^−^/SO_4_^2−^ and F^−^/SO_4_^2−^systems. Due to the differences in Gibbs free energy of hydration among NO_3_^−^, Br^−^, Cl^−^, F^−^, and SO_4_^2−^ ions (Table 1), the permselectivity of NO_3_^−^/SO_4_^2−^, Br^−^/SO_4_^2−^, Cl^−^/SO_4_^2−^, and F/SO_4_^2−^, which decreases with the magnitude of difference in Gibbs free energy of hydration, is 18.4, 17.9, 15, and 5.2, respectively. In particular, the ion flux and permselectivity of the QDPAB-C5 membrane in NO_3_^−^/SO_4_^2−^, Br^−^/SO_4_^2−^, Cl^−^/SO_4_^2−^, and F/SO_4_^2−^ systems are significantly higher than those of commercial Neosepta ACS membranes (Figure 4c). This shows that the QDPAB-C5 membrane can be applied to various monovalent/divalent anion mixed systems. In addition, the ion flux and permselectivity of the QDPAB-C5 membrane in NO_3_^−^/Cl^−^, Br^−^/Cl^−^, and F^−^/Cl^−^ monovalent anion systems were also evaluated. Based on the reasons similar to separating monovalent/divalent anion systems, the QDPAB-C5 membrane is capable of separating diverse monovalent anions due to the difference in Gibbs free energies of hydration (Table 1). As shown in Figure 4d, the permselectivity of NO_3_^−^/Cl^−^, Br^−^/Cl^−^, and F^−^/Cl^−^ is 2.6, 1.7, and 0.32, respectively, showing that the QDPAB-C5 membrane is capable of separating diverse monovalent anions.

## 4. Conclusions

In this research, a series of poly(alkyl-biphenyl pyridinium)-based anion exchange membranes with diverse alkyl side chains were synthesized. Combining the performance of selectivity and area resistance, the QDPAB-C5 membrane has the best performance. Compared to commercial Neosepta ACS membranes, the ion flux and permselectivity of several monovalent/divalent anion separation systems (F^−^/SO_4_^2−^, Br^−^/SO_4_^2−^, NO_3_^−^/SO_4_^2−^, and Cl^−^/SO_4_^2−^) are significantly greater. In addition, the QDPAB-C5 membrane also has a certain degree of permselectivity among several monovalent anion systems (NO_3_^−^/Cl^−^, Br^−^/Cl^−^, and F^−^/Cl^−^). In summary, a permselective anion exchange membrane was generated via the Menshutkin reaction synthesis approach through the innovative design of the polymer’s main chain material and structure. The QDPAB-C5 membrane has broad commercial application prospects in diverse anion systems, and its direct synthesis method may enable it to be commercially prepared. 

## Figures and Tables

**Figure 1 membranes-13-00188-f001:**
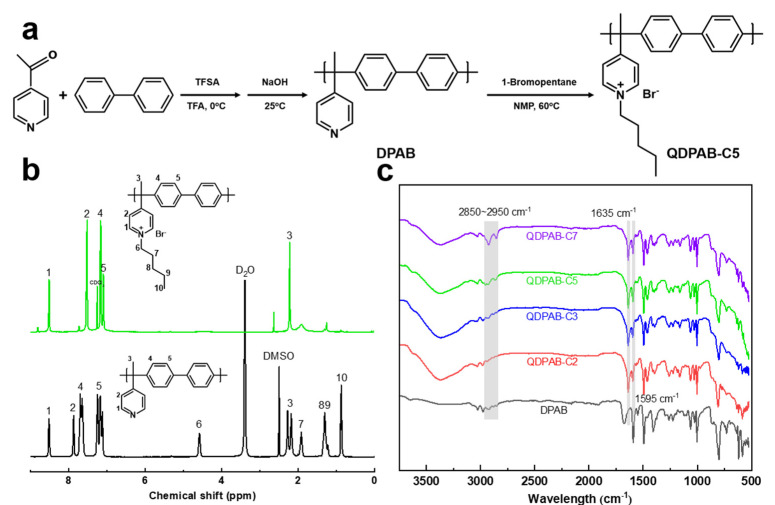
(**a**) The synthesis route of DPAB and QDPAB-C5 polymers; (**b**) ^1^H NMR spectra of DPAB and QDPAB-C5 polymers; (**c**) ATR-FTIR spectra of DPAB and QDPAB-Cx membranes.

**Figure 2 membranes-13-00188-f002:**
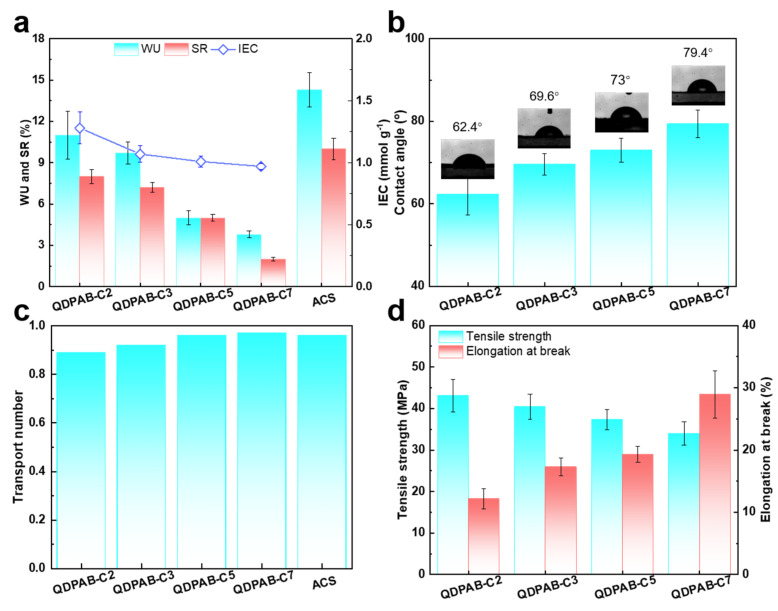
(**a**) IEC, WU, and SR of QDPAB-Cx and ACS membranes; (**b**) CA of QDPAB-Cx membranes; (**c**) transport numbers of QDPAB-Cx and ACS membranes; (**d**) tensile strength and elongation at break of QDPAB-Cx membranes.

**Figure 3 membranes-13-00188-f003:**
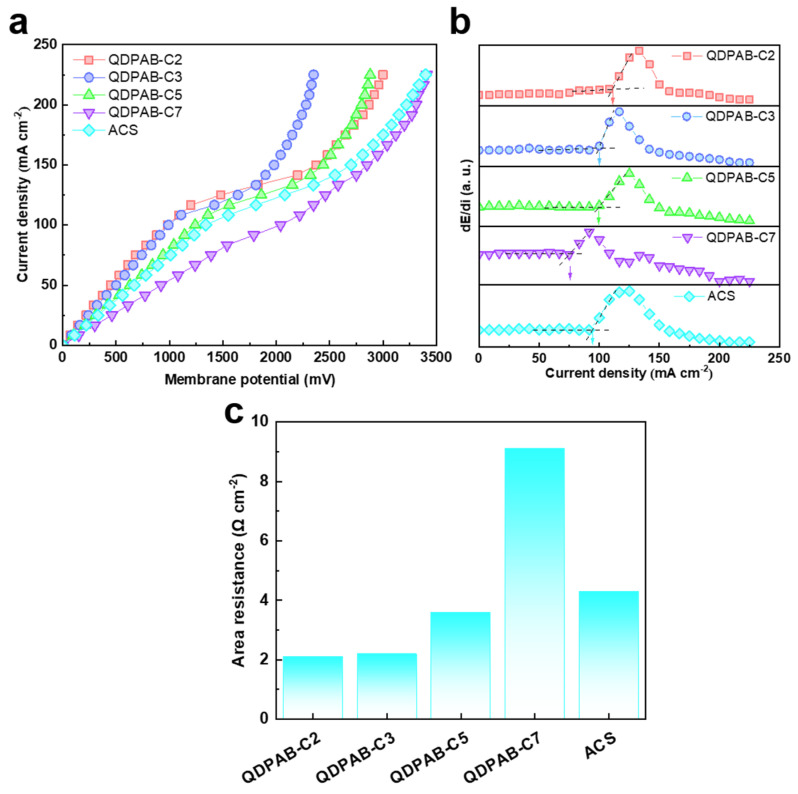
(**a**) Current–voltage curves of QDPAB-Cx and ACS membranes; (**b**) the derivative of dE/di as a function of current density of QDPAB-Cx and ACS membranes calculated from the current–voltage curves; (**c**) the area resistance of the QDPAB-Cx and ACS membranes.

**Figure 4 membranes-13-00188-f004:**
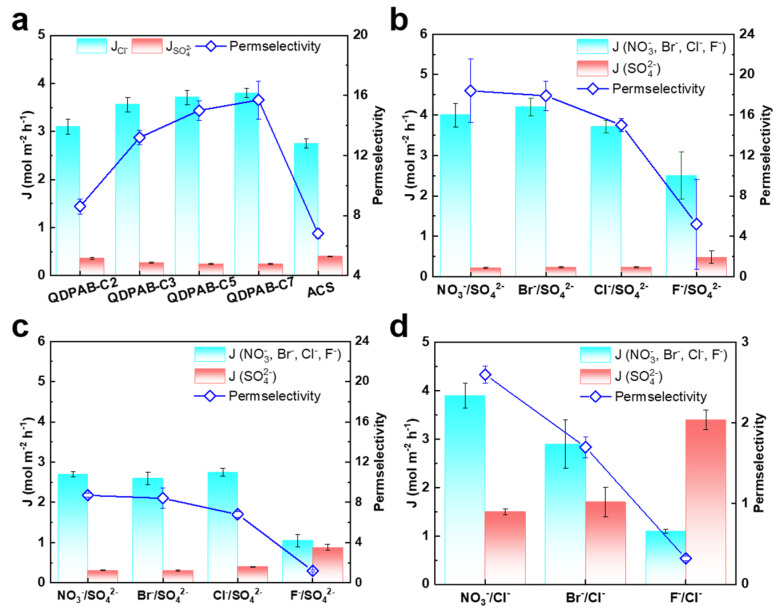
(**a**) Anion flux and permselectivity of the QDPAB-Cx and ACS membranes in a 0.1 mol L^−1^ NaCl/0.1 mol L^−1^ Na_2_SO_4_ mixed solution at a current density of 10 mA cm^−2^; (**b**) anion flux and permselectivity of the QDPAB-C5 membrane in different mixed solutions consisting of various monovalent anions and SO_4_^2−^ ions (0.1 mol L^−1^ NaNO_3_/0.1 mol L^−1^ Na_2_SO_4_, 0.1 mol L^−1^ NaBr/0.1 mol L^−1^ Na_2_SO_4_, 0.1 mol L^−1^ NaCl/0.1 mol L^−1^ Na_2_SO_4_, and 0.1 mol L^−1^ NaF/0.1 mol L^−1^ Na_2_SO_4_) at a current density of 10 mA cm^−2^; (**c**) anion flux and permselectivity of the ACS membrane in different mixed solutions consisting of various monovalent anions and SO_4_^2−^ ions (0.1 mol L^−1^ NaNO_3_/0.1 mol L^−1^ Na_2_SO_4_, 0.1 mol L^−1^ NaBr/0.1 mol L^−1^ Na_2_SO_4_, 0.1 mol L^−1^ NaCl/0.1 mol L^−1^ Na_2_SO_4_, and 0.1 mol L^−1^ NaF/0.1 mol L^−1^ Na_2_SO_4_) at a current density of 10 mA cm^−2^; (**d**) anion flux and permselectivity of the QDPAB-C5 membrane in different mixed solutions consisting of various monovalent anions (0.1 mol L^−1^ NaNO_3_/0.1 mol L^−1^ NaCl, 0.1 mol L^−1^ NaBr/0.1 mol L^−1^ NaCl, and 0.1 mol L^−1^ NaF/0.1 mol L^−1^ NaCl) at a current density of 10 mA cm^−2^.

**Table 1 membranes-13-00188-t001:** Hydrated radii ($r_{h}$ ) and Gibbs free energy of hydration (−Δ$G_{h}^{0}$ ) of I^−^, NO_3_^−^, Br^−^, Cl^−^, F^−^, and SO_4_^2−^ [34].

Anion Species	$-\Delta G_{h}^{0}(\mathrm{kJ}/\mathrm{mol})$	$r_{h}(Å)$
I^−^	257	3.31
NO_3_^−^	270	3.35
Br^−^	303	3.30
Cl^−^	317	3.32
F^−^	434	3.52
SO_4_^2−^	1000	3.79

## Data Availability

The data presented in this study are available on request from the corresponding author.

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
