# Peer review of "Poly(alkyl-biphenyl pyridinium)-Based Anion Exchange Membranes with Alkyl Side Chains Enable High Anion Permselectivity and Monovalent Ion Flux"

_membranes, 2023, doi:10.3390/membranes13020188_

Round 1

Reviewer 1 Report

I think it is of great interest in the community of membranes and their applications. As a result, I will recommend the publication of this manuscript to accept it followed by some minor corrections.

Comments.

1.    Please check English in overall manuscript.

2.    Please change the color of Figure 2 and Figure 4. As the current color doesn’t seems appropriate.

3.    It will be better if Author can plot a area resistance in a graph format. Further I will suggest to show the Nyquist plot for the prepared membrane.

Reviewer 2 Report

In this work, the authors present a poly(alkyl-biphenyl pyridinium)-based anion exchange membranes with alkyl side chains for permselective anion separation. The as-prepared AEM shows the superior separation performance (ion flux and perm-selectivity), relative to commercial Neosepta ACS membrane. I think this work is also an impressive study related to the optimized AEM having certain perm-selectivity among several mono-valent anion systems (NO3-/Cl-, Br-/Cl-, and F-/Cl-). I recommend considering its publication in Membranes, after minor revisions by addressing some concerns listed as below:

(1) The reason why the as-prepared AEM showing the capacity of separating diverse mono-valent anions should be also included.

(2) The references including the data in Table 2 should be listed.
